



# A Framework for Simultaneous Design of Wind Turbines and Cable Layout in Offshore Wind

Juan-Andrés Pérez-Rúa[1] and Nicolaos A. Cutululis[1]

[1]DTU Wind Energy, Technical University of Denmark, Frederiksborgvej 399, 4000 Roskilde, Denmark

**Correspondence:** Juan-Andrés Pérez-Rúa (juru@dtu.dk)

**Abstract.** An optimization framework for simultaneous design of wind turbines (WTs) and cable layout for collection system of offshore wind farms (OWFs) is presented in this paper. The typical approach used in both research and practical design is sequential, with an initial annual energy production (AEP) maximization, followed then by the collection system design. The sequential approach is robust and effective, however it fails to exploit the synergies between optimization blocks. Intuitively, one of the strongest trade-offs is between the WTs and cable layout, as they generally compete, i.e. spreading out WTs mitigates wake losses for larger AEP, but also results in longer submarine cables in the collection system and higher costs. The proposed optimization framework implements a gradient-free optimization algorithm to smartly move the WTs within the project area subject to minimum distance constraint, while a fast heuristic algorithm is called in every function evaluation in order to calculate a cost estimation of the cable layout. In a final stage, a refined cable layout design is obtained by iteratively solving a mixed integer linear program (MILP), modelling all typical engineering constraints of this particular problem. A comprehensive performance analysis of the cost estimation from the fast heuristic algorithm with respect to the exact model is carried out. The applicability of the method is illustrated through a large-scale real-world case study. Results shows that: (i) the quality of the cable layout estimation is strongly dependent on the separation between WTs, where dense WTs layouts present better performance parameters in terms of error, correlation and computing time, and (ii) the proposed simultaneous design approach provides up to 6% of improvement on the quality of fully feasible wind farm designs, and broadly, a statistically significant enhancement is ensured in spite of the stochasticity of the optimization algorithm.

## 1 Introduction

Offshore wind currently represents one of the main drivers towards power systems fully based on renewable energies. The rapid escalation of this technology has been evident in the last decade (GWEC, 2018), where from 2011 to 2019 the global installed capacity multiplied by a factor of seven, reaching the total of 29 GW worldwide as per the last official consolidated report (GWEC, 2020a). The projected compound annual growth rate of the global offshore wind market is of at least 8% for the starting decade, where yearly new installations are expected to surpass the figure of 20 GW (GWEC, 2020b). In the long term, the target set by the European Commission sees offshore wind providing 450 GW by 2050 (BVG Associates et al., 2019). The previous figures are telling of the development maturity of offshore wind industry, however certainly the true global expansion has just started.



The massive global proliferation of offshore wind is backed up by the significant developments of wind turbine (WTs) technology, especially in terms of greater nominal power, enlarged production chain capacities, and application of modern installation practices (WindEurope, 2019). In general, projects are today much larger than a few years ago, in the range of 800 MW, with hundreds of WTs rated $6\,\text{MW}$-$8\,\text{MW}$ WindEurope (2020). Apart from the WTs, the set of supporting and auxiliary

components of the project (balance of plant, BoP) must be sized as well. This mainly includes submarine cables, offshore transformers, converters stations in case of direct current technology (DC), foundations, structures, and protection/control equipment. WTs alone account today for around 30% of the overall levelized cost of energy (LCoE), while a similar figure is attributable to the BoP, with submarine cables particularly representing the most expensive component with around 11% of the LCoE (ORE Catapult, 2020).

As a consequence of economies of scale, the LCoE has considerably decreased in the last decade. However, electrical integration system costs (capital expenses associated primarily to submarine cables and offshore substation) are not following the overall decreasing trend, mainly due to the lack of optimization techniques for designing this subsystem, and the need of longer and bigger submarine cables (Tennet, 2019).

Designing an offshore wind farm (OWF) is a complex and cross-disciplinary scientific task. Given the nature of the technol-

ogy, a vast amount of different disciplines and aspects (or optimization blocks) are involved, theoretically fostering exploitation of the synergies (and trade-offs) between them for the benefit of the system.

Wind farm design optimization is a rich research field that has evolved significantly over the last two decades (Ning et al., 2019; Herbert-Acero et al., 2014). Historically, both research and industry methods focused on maximizing the annual energy production (AEP), though more recently the focus has shifted to minimizing the LCoE accounting for key cost drivers as well.[1]

Wind farm optimization addresses the key elements of overall system design and their impacts on the major LCOE elements:

  – AEP: AEP is driven by the size of the overall turbines (rated power, rotor diameter) and their performance characteristics, as well as their interaction with each other through WTs wakes that reduce production from downstream machines. Optimization of the turbine sizing, selection, number and layout have all been considered for the purposes of maximizing AEP. This problem has been studied broadly and intensively. The pioneering work of Mosetti et.al. (Mosetti et al., 1994)

marked the initial endeavour, applying a genetic algorithm and implementing a similar version of the Katic-Jensen wake decay model (Katic et al., 1986). Plethora of works followed, in general proposing optimization heuristics divided into two categories: 1) Gradient-free (heuristics and metaheuristics), such as in (Marmidis et al., 2008) (monte carlo), (Grady et al., 2005) (genetic algorithm), (Wan et al., 2010) (particle swarm optimization), (Wagner et al., 2013) (local search), among others; and 2) Gradient-based as in (Thomas and Ning, 2018; Stanley and Ning, 2019a; Brogna et al., 2020).

---

[1]The basic equation for LCoE is:

$$LCoE = (FCR \cdot CAPEX + OPEX)/AEP \qquad (1)$$

where FCR is the fixed-charge rate, CAPEX are the capital expenditures, and OPEX are the operational expenditures. See (Dykes et al., 2017) for a detailed breakdown of LCOE calculations.



- CAPEX for the BoP: BoP optimization is often done as a sub-optimization problem of the electrical system (a key cost component), though more detailed optimization has also been extensively studied (as will be discussed). Other costs for BoS such as the roads for land-based farms have been explored and even the installation and logistics strategies (Roscher et al., 2020).

- CAPEX for WTs and foundations: While typically, the WT itself is taken as fixed, there has been more recent research that even looks at integrated design of the WT and the wind farm (Stanley et al., 2018; Stanley and Ning, 2019c, a; Graf et al., 2016). In addition, for offshore sites with varied sea-depths and soil conditions, the design of support structures have been considered either as a sub-optimization or in a sequential optimization (Sanchez Perez-Moreno et al., 2018).

- OPEX: Reducing OPEX and ensuring site suitability of the WTs for a given layout is a more recent research topic that typically requires using surrogate models to include load models in the optimization (Riva et al., 2020).

While all of these elements are important to full LCoE analysis and optimization of OWFs, one of the strongest trade-offs from overall cost metrics perspective is the placement of WTs and the total length of submarine cables: the more spread out the WTs, the less flow interaction between them -and more AEP-, at the expense of larger cable length. Given the theoretical strong dependency and interrelation between them, a coordinated design should bring the best outcome for the whole system.

However, on its own, the design of electrical systems for offshore wind define a set of highly complex optimization problems. Among them, the cable layout problem for collection systems is defined as how to interconnect the fixed-positioned WTs towards the Offshore Substations (OSSs), given a set of available cables and engineering constraints (no crossing of cables, maximum number of main feeders, topology, etc.). The cable layout problem maps to standard computer science problems categorized as NP-Hard (Pérez-Rúa and Cutululis, 2019), implying the lack of efficient methods (polynomial running time algorithms), greatly affecting the tractability for large-scale projects. Proposed methods to approach this problem can be categorized as: 1) Heuristics (Hou et al., 2016a; Pérez-Rúa et al., 2019a), 2) Metaheuristics (Hou et al., 2016b; Minguijón et al., 2019), and 3) Global optimization (Fischetti and Pisinger, 2018; Pérez-Rúa et al., 2019b), where mixed integer linear programming (MILP) is the most used formulation. A survey and analysis of these methods can be found in (Lumbreras and Ramos, 2013; Pérez-Rúa and Cutululis, 2019).

As mentioned before, the a priori strong dependency between WT layout and collector system optimization is a solid motivation to research on simultaneous optimization, instead of the classical sequential approach popularly used in the industry (Pérez-Rúa and Cutululis, 2019) (See Fig. 1). The main challenge is developing formulations and methods with good properties: feasibility, numerical tractability, efficiency, effectiveness, and (reduced) complexity for implementation.

A few efforts towards simultaneous optimization of WTs layout using gradient-free methods and implementing as cost component heuristic algorithms for the cable layout have been identified in (Sanchez Perez-Moreno et al., 2018; Wade et al., 2019; Amaral and Castro, 2017). These algorithms are in general fast, but their calculated cost may differ considerably from the global optimum of the cable layout cost function. Most importantly, feasible designs are not guaranteed using heuristics, therefore still needing a final stage to accurately optimize the cable layout (not addressed in the mentioned literature). Finite grid-based WTs layout have been pre-defined and the cable layout optimized for each of them using MILP in (Marge et al.,





2019); this procedure allows for constructing spatial regression functions to estimate cost functions, however the search space

is artificially biased towards a specific topology. A bi-level (nested) multi-objective optimization framework has been proposed
in (Tao et al., 2021), to address WTs layout, cable layout, interaction with the power grid, and power quality, implementing a
particle swarm optimization algorithm. Wind turbines and cable layout are solved separately with this metaheuristic algorithm,
which potentially would imply a significant problem regarding tractability for large-scale instances. A similar method has been
proposed in (Wu et al., 2014), where an ant colony system concept is designed and implemented to design the cable layout,

relaxing the model by neglecting typical engineering constraints.

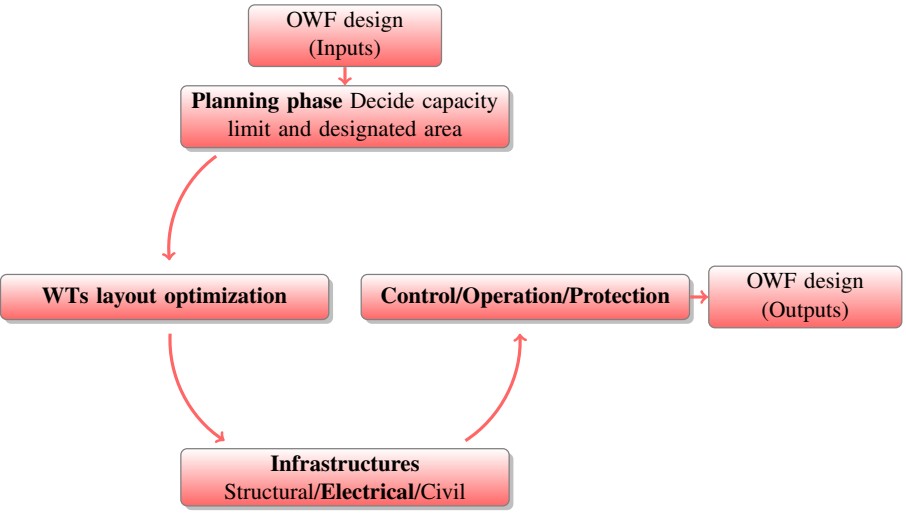

**Figure 1.** Industrial approach for OWF design (Pérez-Rúa and Cutululis, 2019).

In this sense, the main contribution of this article is the proposition of a framework for simultaneous design of WTs and
cable layout, where a feasible design is obtained, in contrast to previous works (Sanchez Perez-Moreno et al., 2018; Wade et al.,
2019). This framework is benchmarked against the sequential approach commonly employed in the industry. The comparison is
elaborated in terms of feasible design using high-level economic metrics. Aiming to study the underlying operating principles

of the proposed framework, a systematic investigation of the performance of the heuristic for cable layout optimization is
performed, having as reference the exact optimization model for the cable layout design (Pérez-Rúa et al., 2019b). Likewise, a
sensitivity analysis for different OSS position is presented, in order to quantify the effect over the performance comparison.

## 2 Methods

### 2.1 Engineering wind farm model, AEP calculation, and WTs layout design

The engineering wind farm model implemented in this manuscript is the so-called IEA-37 Simple Bastankhah Gaussian avail-
able in PyWake (Pedersen et al., 2019). This model is composed by: (i) a component to propagate the wake in the wind farm,





performing a minimum of deficit calculations, by studying the effect of a particular WT over its downstream WTs. This procedure neglects upstream blockage effects but is computationally fast. (ii) a component to compute the previously mentioned wake velocity deficits between pairs of WTs, and (iii) a component to superpose the deficits for turbines placed in multiple

wakes to obtain the total velocity deficit through a squared sum operation. Formulae of this model is available in (IEA Wind Task 37, 2019).

Let $\boldsymbol{X}$ and $\boldsymbol{Y}$ be a set of abscissas and ordinates, respectively, of the $n_{\mathrm{w}}$ WTs to install for the OWF to be optimized. The AEP for this particular layout is calculated taking into consideration the WT parameters (most importantly the power curve), the number of wind directional bits (fixed to 16 in this paper), and the site's wind rose. A valid layout represented by $\boldsymbol{X}$ and

$\boldsymbol{Y}$ must satisfy two basic constraints: (i) WTs must be placed inside of the polygon that defines the OWF designated area, and (ii) a minimum distance of $2D$ between WTs, where $D$ is the WT diameter, must be guaranteed.

### 2.2  Collection system cable layout

The aim is to design the cable layout of the collection system for an OWF, i.e., to interconnect the $n_{\mathrm{w}}$ WTs to the available OSSs, $n_{\mathrm{o}}$, using a list $\boldsymbol{T}$ of cables available, minimizing the total investment cost. The collection system cable layout optimization is

represented as a static problem with respect to time, with the nominal power being generated by the WTs. This is to ensure robustness on the design given the uncertainty associated to real time power profile. For simplicity, only one OSS is considered, hence $n_{\mathrm{o}} = 1$. Let the WTs define the sets, $\boldsymbol{N}_{\mathrm{w}} = \{2, \cdots, 1 + n_w\}$, and $\boldsymbol{N} = 1 \cup \boldsymbol{N}_{\mathrm{w}}$. The physical locations of points in $\boldsymbol{N}$ are correspondingly available in $\boldsymbol{S}$ as abscissas ($x_1$ for OSS and $\boldsymbol{X}$ for $\boldsymbol{N}_{\mathrm{w}}$) and ordinates ($y_1$ for OSS and $\boldsymbol{Y}$ for $\boldsymbol{N}_{\mathrm{w}}$) in the two-dimensional space. The Euclidean distance between the positions of the points $i \in \boldsymbol{N}$ and $j \in \boldsymbol{N}$, is denoted as $d_{ij}$. The

complete weighted directed graph $G(\boldsymbol{N}, \boldsymbol{A}, \boldsymbol{D})$ gathers all relevant graph-related parameters, where $\boldsymbol{N}$ represents the vertex set, $\boldsymbol{A}$ the set of arcs arranged as a pair-set $a \in \boldsymbol{A} : a = (i, j)$, and $\boldsymbol{D}$ the set of distances $d_a$.

Regarding cables, let the capacity of a cable $t \in \boldsymbol{T}$ be $u_t$ measured in terms of number of WTs connected downstream. Hence, let $\boldsymbol{U}$ be the set of capacities sorted as in $\boldsymbol{T}$. Each cable type $t$ has a cost per unit of length, $c_t \in \boldsymbol{C}$, in such a way that $\boldsymbol{T}, \boldsymbol{U}$, and $\boldsymbol{C}$ are all comonotonic.

Generally, a feasible collection system design includes the following engineering constraints:

**[C1]** A tree topology must be enforced. This means that there must be only one electrical path from each WT towards the OSS.

**[C2]** The capacity of cables must not be exceeded.

**[C3]** The number of main feeders, i.e., cables reaching directly the OSS, must be limited to a maximum $\phi$.

**[C4]** Cables must not lay over each other (no crossing cables) due to practical installation aspects.

As explained in the introduction, the formally elucidated problem above can be tackled with methods classified into three groups: heuristics, metaheuristics, and global optimization. Section 2.2.1 focuses on a specific class of heuristics (two-steps), and Sect. 2.2.2 concentrates on an exact model based on a MILP formulation which brings along optimality certificate. This



allows assesing how far away is the best known solution to the best known achievable objective value computed as a percentage
gap. Both methods are applied in the proposed optimization framework. Metaheuristics are excluded from this work as their
value is usually utilized when stand-alone solver-free robust methods are required for the cable layout optimization problem.
It is considered a cumbersome endeavour to harmonize metaheuristic methods for both the cable and WTs layout problem in
either a nested or fully integrated fashion.

### 2.2.1 Heuristics optimization algorithms

Heuristics are defined in this application as solver-free polynomial running time algorithms, that through a set of sequential
steps, construct a (hopefully) feasible design or infeasible point with an associated cost. In either case, heuristics are useful
in this context for obtaining a very fast estimation of the cost of the cable layout for given WTs and OSS position. This is
required since during the iterative process of simultaneously solving the WTs and cable layout problem, function evaluations
are repeatedly executed to compute the value of the high-level economic metric as part of the process to move throughout the
search space. The faster the heuristics, the more iterations for a given computing time budget, and thus the more exhaustive the
design exploration.

The flowchart of the heuristic implemented in this article is presented in Fig. 2. The algorithm consists of a two-steps
decision process, where the first step selects the arcs that connect WTs to each other and towards the OSS. As illustrated in
Fig. 2, several capacitated minimum spanning tree (C-MST) greedy heuristic algorithms can cope with the first step, such as
Prim, Kruskal, Vogel's approximation method, Esau-Williams, among others. Although these algorithms have been proposed
individually ((Prim, 1957; Kruskal, 1956; Chandy and Russell, 1972; Esau and Williams, 1966)), they intrinsically follow the
same underlying rules during the design construction process (Kershenbaum, 1974; Kershenbaum and Chou, 1974), based on
the generalization of trade-off cost functions updated correspondingly for each case.

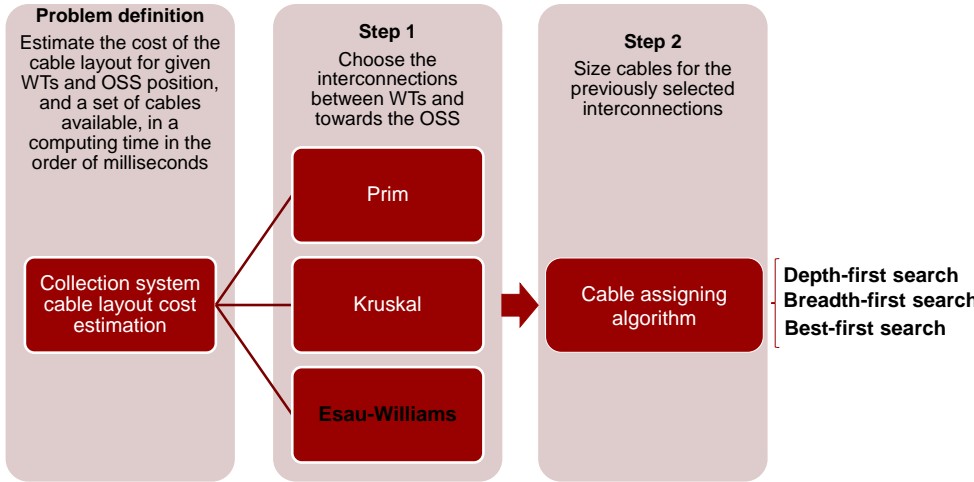

**Figure 2.** Two-steps heuristic for collection system cable layout cost estimation.



The satisfiability of [C1], [C2], [C3], and [C4] is sought in the Step 1. [C1] is fulfilled by means of avoiding the interconnection of elements in $N$ belonging to different components at given iteration, with subsequent updating of disjoint component sets after merging them if all other constraints are not violated. [C2] is guaranteed through the limitation of nodes number to $U = \max U$ of any maximal subgraph connected to the OSS with a single arc. [C3] is enforced by simply controlling the number of arcs reaching the OSS. Finally, [C4] is respected by checking at a given iteration the candidate arc with the cumulative set of selected arcs.

[C1] is deemed as a fundamental constraint, considering that the problem is closely defined to generate trees, and any other topology would lead to very unrealistic cost compared to the global minimum. Computational experiments carried out in a previous work (Pérez-Rúa et al., 2019a) showed that the coexistence of [C2] with [C3] (and by extension [C4]) frequently leads to infeasible points (i.e. a forest graph), as a result of backtracking inability. This means that one needs to prioritize the exclusive evaluation of these constraints. The relaxation of [C2] could harden the calculation of the cable layout cost estimation, considering that the cost vector $C$ is bounded to the available set of cables $T$. Consequently, the reasoning to disregard [C3] and [C4] is twofold. First, the computational aspects related to the evaluation of these constraints are avoided, reducing the computing time for every function evaluation. Second, as the nature of these heuristics overestimate the cost of the cable layout (due to the two-step process), the relaxation of [C3] and [C4] could help decrease the gap with respect to the global minimum.

Ultimately, Esau-Williams algorithm has consistently performed better in terms of feasible points and investment cost quality, when framed in the method of Fig. 2, than the other greedy heuristics (Pérez-Rúa et al., 2019a). On that account, a modified version of Esau-Williams heuristic is solely implemented in this work. The pseudocode for Step 1 is presented in Algorithm 1.

The first five lines initialize the useful sets. The most important is the trade-off set $T_o$ using the weight parameters $p_i$. Note that according to the mathematical definition, priority is given to the arcs located the farthest to the OSS. The iteration process starts at line six and continues until a fully connected tree graph is obtained. In each iteration the arc with the lowest trade-off value is incorporated to the tree, as long as it satisfies [C1] and [C2]. In that case, the component sets are merged, and the trade-off values linked to the newly formed component are updated. Otherwise, the arc $a_o$ and its inverse $\bar{a}_o$ become completely banned from the design process, by equalizing their trade-off values to infinite. The indirect graph $G_d$ in line 19 contains the graph tree nullifying any directionality of arcs.

In Step 2, any algorithm for traversing or searching $G_d$ can be applied with the purpose of determining the number of WTs connected downstream in each edge $e \in E_o$. Efficient algorithms are, for instance, Depth-first search (Tarjan, 1972), Breadth-first search (Zhou and Hansen, 2006), and Best-first search (Dechter and Pearl, 1985).

Let $\beta$ be the number of WTs connected through edge $e \in E_o$ with length $d_e$. The following trivial optimization problem must be solved: $\min\{X_c^\intercal \cdot C \cdot d_e : X_c^\intercal \cdot U \le \beta, \|X_c\|_1 = 1, X_c \in \mathbb{B}^{|T|}\}$. The solution provides the cheapest cable $t \in T$ able to support $\beta$ WTs via $x_{c_t} \in X_c$, where each tuple $(e \in E_o, \beta)$ defines an independent problem solved in linear running time.



---

**Algorithm 1** Step 1 of Fig. 2

---

1: $\forall i \in \boldsymbol{N}, p_i \leftarrow d_{i1}$

2: $\forall a \in \boldsymbol{A}, t_a \leftarrow d_{ij} - p_i : t_a \in \boldsymbol{T_o}$

3: $\forall i \in \boldsymbol{N}, \boldsymbol{C}_{oi} \leftarrow i$

4: $\boldsymbol{E}_o \leftarrow \emptyset$

5: $\boldsymbol{D}_o \leftarrow \emptyset$

6: **while** $(\forall i \in \boldsymbol{N}, \boldsymbol{C}_{oi} \neq \boldsymbol{N}) \wedge (\forall t_a \in \boldsymbol{T}_o, t_a \neq Inf)$ **do**

7:     $t_o \leftarrow \min \boldsymbol{T}_o$

8:     $a_o \leftarrow \arg\min \boldsymbol{T}_o : a_o = (i,j)$

9:     **if** *Satisfied [C1] $\wedge$ [C2]* **then**

10:         $\boldsymbol{E}_o \leftarrow \boldsymbol{E}_o \cup a_o$

11:         $\boldsymbol{D}_o \leftarrow \boldsymbol{D}_o \cup d_{a_o}$

12:         $\forall i \in \boldsymbol{C}_{oi}, p_i \leftarrow p_j$

13:         $\forall a \in \boldsymbol{A}, t_a \leftarrow d_{ij} - p_i$

14:         $\boldsymbol{C}_{oi}, \boldsymbol{C}_{oj} \leftarrow \boldsymbol{C}_{oi} \cup \boldsymbol{C}_{oj}$

15:     **end if**

16:     $t_{a_o} \leftarrow inf$

17:     $t_{\bar{a}_o} \leftarrow inf$

18: **end while**

19: *Output indirect graph $G_d(\boldsymbol{N}, \boldsymbol{E}_o, \boldsymbol{D}_o)$*

---

### 2.2.2 Global optimization: MILP model

When a problem is formulated within the framework of a MILP paradigm, state-of-the-art solvers can be applied in quest of high-quality solutions with proved optimality certificate. MILP models are generally more efficient than mixed integer quadratic programming (MIQP), and mixed integer non-linear programming (MINLP), therefore they are the preferred choice for the nature of this problem. A challenge is then how to incorporate real-world engineering aspects into the optimization, such as quadratic power losses or other type of non-linearities. In the following, a basic model, stemming from (Pérez-Rúa et al., 2019b), with the goal to design the cable layout to minimize total investment, while satisfying [C1] to [C4] is deployed.

$$x_{ij} \in \{0,1\} \quad y_{ij}^k \in \{0,1\} \quad \forall (i,j) \in \boldsymbol{A} \wedge k \in \{1, \cdots, f(i)\} \tag{2}$$

$$\min \sum_{i \in \boldsymbol{N}} \sum_{j \in \boldsymbol{N}_{\mathrm{w}}} \sum_{k=1}^{f(i)} c_{ij}^k \cdot y_{ij}^k \tag{3}$$





$$\sum_{i \in \boldsymbol{N}} \sum_{k=1}^{f(i)} y_{ij}^k = 1 \quad \forall j \in \boldsymbol{N}_{\mathrm{w}} \tag{4}$$

$$\sum_{i \in \boldsymbol{N}} \sum_{k=1}^{f(i)} k \cdot y_{ij}^k - \sum_{i \in \boldsymbol{N}_{\mathrm{w}}} \sum_{k=1}^{f(i)} k \cdot y_{ji}^k = 1 \quad \forall j \in \boldsymbol{N}_{\mathrm{w}} \tag{5}$$

$$\sum_{j \in \boldsymbol{N}_{\mathrm{w}}} \sum_{k=1}^{f(1)} y_{1j}^k \le \phi \tag{6}$$

$$x_{ij} + x_{ji} + x_{uv} + x_{vu} \le 1 \quad \forall \{(i,j),(u,v)\} \in \boldsymbol{\chi} \tag{7}$$

$$\sum_{k=1}^{f(i)} y_{ij}^k - x_{ij} \le 0 \quad \forall (i,j) \in \boldsymbol{A} \tag{8}$$

$$-\sum_{i \in \boldsymbol{N}} \sum_{k=v+1}^{f(i)} \left\lfloor \frac{k-1}{v} \right\rfloor \cdot y_{ij}^k + \sum_{i \in \boldsymbol{N}_{\mathrm{w}}} \sum_{k=v}^{f(i)} y_{ji}^k \le 0 \quad \forall v = \{2, \cdots, U-1\} \wedge j \in \boldsymbol{N}_{\mathrm{w}} \tag{9}$$

Eq. (2) defines the variables of the model. Binary variable $x_{ij}$ is one if the arc $(i,j)$ is selected in the solution, and zero otherwise. Likewise, binary variable $y_{ij}^k$ models the $k$ number of WTs connected downstream from $j$, including the WT at node $j$ (under the condition that $x_{ij} = 1$). Function $f(i)$ maps from tail node $i$ to maximum number of WTs connectable $k_{\mathrm{m}}$ through an arc $(i,j)$. If $i = 1$ (i.e. the OSS), then $k_{\mathrm{m}} = f(i) = U$, otherwise $k_{\mathrm{m}} = f(i) = U - 1$.

Eq. (3) is the objective function. Cost parameter $c_{ij}^k$ encodes the optimum cost to connect $k$ WTs through arc $(i,j)$ and is obtained similarly to Step 2 in Fig. 2. [C2] is enforced at this point as well. Eq. (4) ensures simultaneously a tree topology, only one cable type used per arc, and the head-tail convention, while Eq. (5) is the flow conservation which avoids forest graph; both Eq. (4) and Eq. (5) guarantee [C1]. Eq. (6) expresses the requirement of [C3]. The set $\chi$ stores pairs of arcs $\{(i,j),(u,v)\}$, which are crossing each other. Excluding crossing arcs ([C4]) in the solution is ensured by the simultaneous application of Eq. (7) and Eq. (8). Finally, Eq. (9) defines a set of valid inequalities to tighten the mathematical model.

The MILP model from Eq. (2) to Eq. (9) presents less variables and constraints than flow-based formulations (Pérez-Rúa et al., 2019b). The latter, in practical terms, facilitates the convergence of solvers in the matter of computing time and memory. Nevertheless, computational limitations are still conspicuous when solely applying the full model using state-of-the-art solvers, as for instance the branch-and-cut by CPLEX (IBM, 2021). As a way around this difficulty, an algorithmic framework wraps up the MILP model as a way around of this difficulty, as shown in Fig. 3.



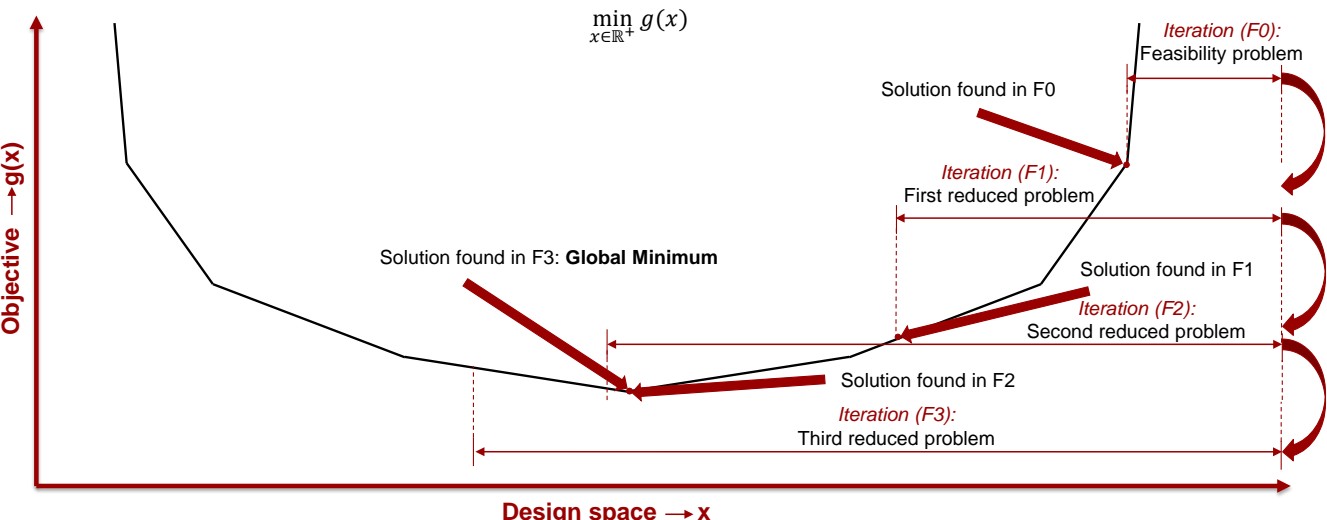

**Figure 3.** Global optimization model: Combination of MILP and heuristics.

The algorithmic framework of Fig. 3 consists of a progressive enlargement of the design (search) space from a rather small size (feasibility problem F0) until a larger one (iteration F3) to (hopefully) reach the global minimum. The transition between iterations is given by a step size determined heuristically and defined as the number of candidate arcs from a WT towards neighbouring ones; this implies that in each iteration, model Eq. (2) to Eq. (9) is reformulated, such as $A_{F1} \subset A_{F2} \subset \cdots A_{Fn} \subset A$. Each iteration is warm-started with the incumbent solution for the sake of shortening the processing, resembling a hill-climbing approach. A compromise between computing time and likelihood to cover the global minimum must be accounted for within this optimization model. The stopping criterion is finding the same solution in two consecutive iterations, indicating that that further enlarging the design space is not required, as represented between F2 and F3 in Fig. 3.

## 2.3 Optimization framework

The objective is to design an OWF that will maximixe the internal rate of return (IRR) of the project. The IRR estimates the profitability of potential investments using a percentage value rather than a dollar amount (Investopedia, 2021), making it preferable by investors in many situations. IRR is often in agreement with other capital budget evaluation metrics as net present value (NPV). The higher the value of IRR, the better a project from the investor's perspective, as it can cover higher values of weighted average cost of capital (WACC). The implemented IRR model (DTU Wind Energy, 2021) is comprehensive, and takes into account a wide range of costs, like DEVEX (costs spent in the period from idea to development to design & planning, CAPEX (expenditures in the period of construction up to the date the OWF is commissioned), OPEX (costs in the operational period), and ABEX (costs related to abandonment of the OWF) (Megavind, 2021).

The AEP affects directly the cash flow along the project lifetime, while the collection system cable layout cost is reflected uniquely in the CAPEX. Subsequently, the IRR metric may weight out more the AEP (in form of economic profits) compared





to the LCoE, where the energy production has potentially less share over the metric in form of a single term. The comparison between AEP and cable costs may be then conservative within this framework, and would challenge to a greater extent the benefits of a simultaneous design of these aspects. As per the results of (Sanchez Perez-Moreno et al., 2018), the strongest trade-off is determined between the cables and the WTs layout, which is further studied in this manuscript. Trade-offs between WTs-foundations, and WTs-structures are deemed negligible for sites with uniform seabed profile, and topographically homogeneous available area.


The function to estimate the cost per meter for each element in $C$ is presented in Eq. (10) (Lundberg, 2003). The cost function is scaled to take into consideration macroeconomics phenomena such as inflation and exchange rate.

$$c_t = a_{p_t} + b_{p_t} \cdot e^{\left(\frac{c_{p_t} S_{n_t}}{10^8}\right)^2} \tag{10}$$

where $a_{p_t}$, $b_{p_t}$, and $c_{p_t}$ are coefficients dependant on the nominal voltage of cable type $t \in T$, $S_{n_t}$ is the rated power of $t$ in VA (also depending of the rated line to line voltage level, $V_n$), and $c_t$ the cost of $t$ in €/km. Note the exponential cost trend in function of the rated power.

### 2.3.1 Design approaches

The classical sequential approach is presented in Fig. 4, where the WTs and cable layout optimization are completely decou-
pled. The simultaneous approach presented in this paper is shown in Fig. 5, incorporating an initial cable layout optimization (Fig. 2) in Task1. For both approaches, Task 2 is defining the final cable layout (based on method of Sect. 2.2.2) given the position of WTs and OSS decided in Task 1, $S$. In the end, a feasible design is obtained for both cases, respecting all constraints of Sect. 2.1 and Sect. 2.2, with an associated objective function value to each.

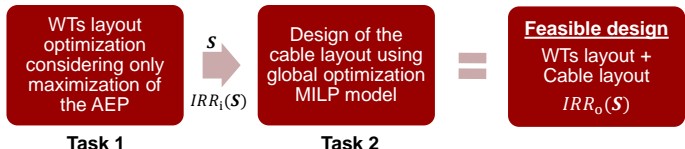

**Figure 4.** Approach 1: Sequential design process for WTs and cable layout.

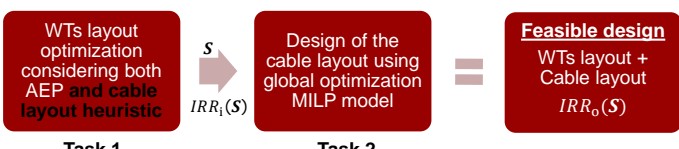

**Figure 5.** Approach 2: Simultaneous design process for WTs and cable layout.




### 2.3.2 Optimization algorithm: Random search

The optimization algorithm for Task 1 of Fig. 4 (sequential, Approach 1) and Fig. 5 (simultaneous, Approach 2) is deployed in Fig. 6. The Random Search algorithm was firstly proposed by (Feng and Shen, 2015) as a tailored-made method for the WTs layout problem using a gradient-free technique. This algorithm has been designed to maximize the AEP, and according to computational experiments, outperforms other gradient-based and gradient-free algorithms for OWFs with 20 to 80 WTs (Feng and Shen, 2015; Brogna et al., 2020). While for Task 1 of Fig. 4 gradient-based algorithms can be implemented, in the

case of a simultaneous design of WTs and cable layout, the intrinsic discontinuous non-smooth nature of the cable layout cost function extremely hardens the computation of analytical or numerical derivatives. As a result of this condition, gradient-free, tree search or local search solvers emerge as good alternatives to tackle a simultaneous design (Hutter et al., 2011).

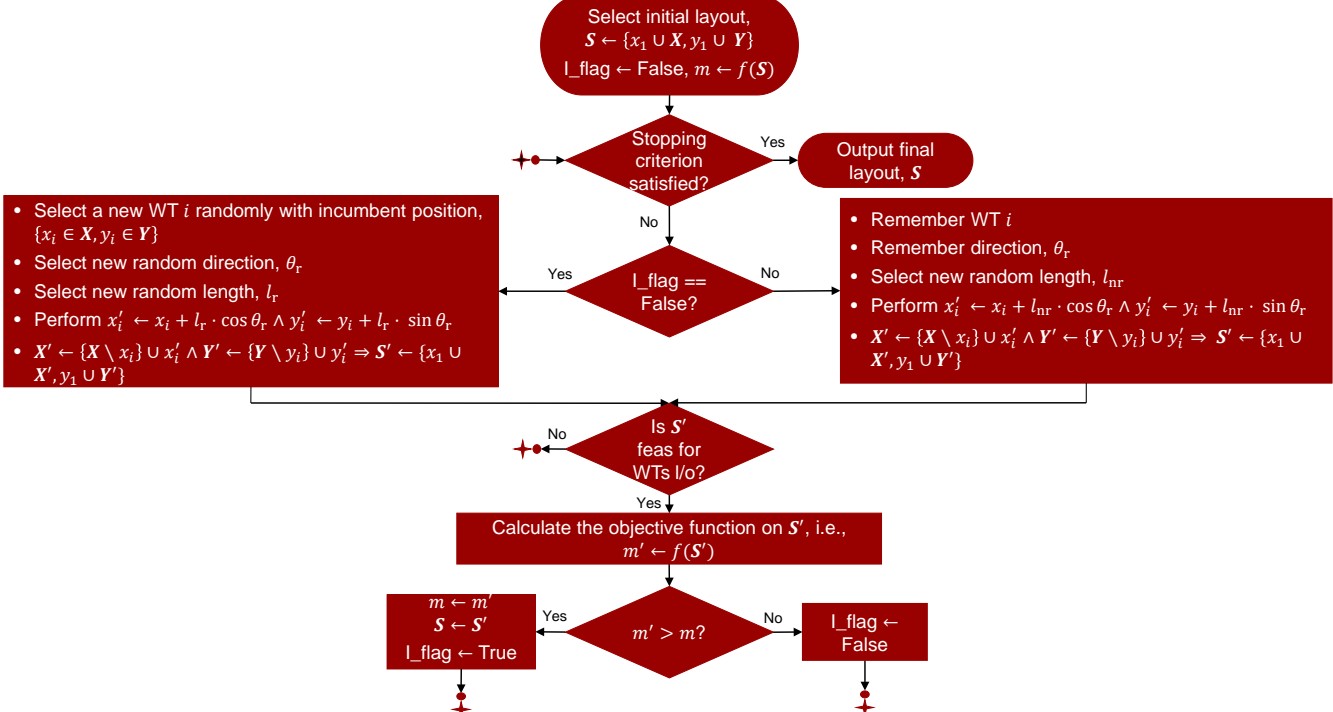

**Figure 6.** Random Search algorithm used for Task 1.

    A generalization of the Random Search algorithm for generic objective function is diagrammed in Fig. 6. The coordinates set, $\boldsymbol{S}$, the improvement flag, I_flag, and the incumbent value of the objective function $m = f(\boldsymbol{S})$ (in this case as mentioned

before, $IRR_i(\boldsymbol{S})$) are initialized. When applied in Approach 1, $m$ assumes a zero cost for the cable layout during the whole iterative process of computing $IRR_i()$. In contrast, when implemented in Approach 2, $m$ considers both the cash flow from the AEP, and capital cost from the cable layout as estimated by the process of Fig. 2, for a given $\boldsymbol{S}$.





Through the output of Task 2 in Fig. 4 and Fig. 5, the overall IRR, $IRR_\mathrm{o}(\boldsymbol{S})$, in each case is computed accounting for the global minimum cost of the cable layout, value that is used in the final comparison analysis.

Since, in theory, the location of the OSS, $\{x_1, y_1\}$, can take infinite values, a course of action is to fix the OSS location in the centroid of the WTs. However, only considering the project area centroid could lead to unrealistic designs, as presented in Fig. 7, where an overlapping between a WT and the OSS appears. In order to avoid this erratic design, the OSS location is displaced to the centroid of the four nearest WTs, in case of the original distance between the points denoted by the OSS and nearest WT is under a specific threshold. Assuming a minimum distance between WTs of $2D$, a minimum distance between

OSS and WT of $\sqrt{2}D$ is ensured after this correction. Consequently, the minimum horizontal distance between the relocated OSS and closest WT's blade tip is $(2\sqrt{2}-1)D/2 \approx 0.91D$, for a threshold distance of $\sqrt{2}D$.

Continuing with the flowchart of Fig. 6, after checking out the stopping criterion (computing time $t_\mathrm{r}$ in this paper), the process carries on with either moving a random WT $i$ across a random direction $\theta_\mathrm{r}$ and length $l_\mathrm{r}$, or continuing with picking up the same WT $i$ as in the previous iteration, moving it towards the same direction $\theta_\mathrm{r}$ with new random length $l_\mathrm{nr}$. A WT

$i$ is shifted towards the same direction $\theta_\mathrm{r}$, if improvement of the objective function has previously been gained in a feasible design in terms of the WTs layout (constraints presented in Sect. 2.1), otherwise a new WT prospect is evaluated as stated befor. Note that the layout initially generated evolves during the running of the algorithm, where the incumbent layout $\boldsymbol{S}$, objective function $m$, and current WT prospect $i$ with direction $\theta_\mathrm{r}$ are memorized, and eventually used for learning purposes in subsequent iterations. This algorithm follows a hill-climbing path with stochasticity incorporated.

The simplicity and rather low computational burden of the algorithm are very important brightsides that ultimately play a major role in its favour concerning tractability and effectiveness, as gradient-free algorithms typically call a great number of function evaluations (Stanley and Ning, 2019b). These properties are notoriously important for large-scale problems ($n_\mathrm{w} > 70$).

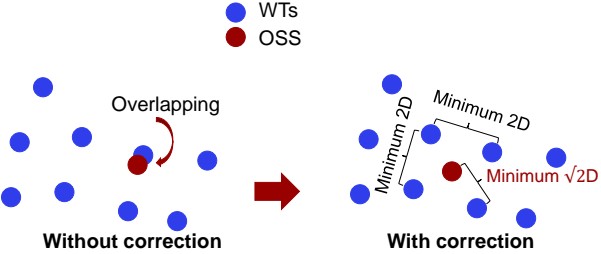

**Figure 7.** Relocation of the OSS at approximately the centroid of the WTs.

## 3 Results

The methodology presented in Sect. 2 is applied to the reference OWF IEA-37 Borssele (Dykes et al., 2015), with the OWF

designated area defined as the convex hull of the polygon vertices. The project consists of 74, $10\,\mathrm{MW}$ each, IEA Task 37 reference WTs (Bortolotti et al., 2015). The main input parameters for the OWF and WT are presented in Table 1 and Table 2, respectively. The short distance to shore ($10\,\mathrm{km}$) permits focusing on the collection system cable layout rather than in the





export system (Pérez-Rúa et al., 2020). In turn, an uniform seabed profile sheds lights on the synergies between WTs and cable layout, as foundations and structures then have an unique design. The energy price of 27 €/MWh corresponds to the minimum

median of the hourly spot prices in the Nordic/Baltic/CWE power markets so far in 2021 (Nord Pool, 2021).

**Table 1.** Parameters of the OWF.

| Offshore Wind Farm | |
| --- | --- |
| **Name** | IEA-37 Borssele site |
| **Number of WTs**, $n_\mathrm{w}$ | 74 |
| **Number of main feeders**, $\phi$ | 10 |
| **Distance from shore** [km] | 10 |
| **Energy price** [€/MWh] | 27 |
| **Project lifetime** [years] | 20 |
| **Uniform water depth** [m] | 20 |

Table 2 deploys the data related to the set of cables available, rated at a voltage level of $V_\mathrm{n} = 33\,\mathrm{kV}$, and their thermal capacity as defined in Sect 2.2; by means of the set $U$ and $V_\mathrm{n}$, the cost set $C$ is calculated implementing Eq. 10. Finally, the site's wind rose in Fig. 8 shows that the prevailing winds come in the sector from 210° to 300° in the Cartesian reference plane.

**Table 2.** Main parameters of the WT.

| Wind Turbine | |
| --- | --- |
| **Reference** | IEA-10MW |
| **Nominal Power** [MW] | 10 |
| **Diameter,** $D$ [m] | 190.6 |
| **Hub height** [m] | 119 |
| **Nominal RPM** | 8.68 |

**Table 3.** Cables available, $T$.

| Cables available | |
| --- | --- |
| $V_\mathrm{n} = 33\,\mathrm{kV}$ | |
| **Cable set,** $T$ | **Capacity set,** $U$ |
| 1 | 4 |
| 2 | 6 |
| 3 | 8 |





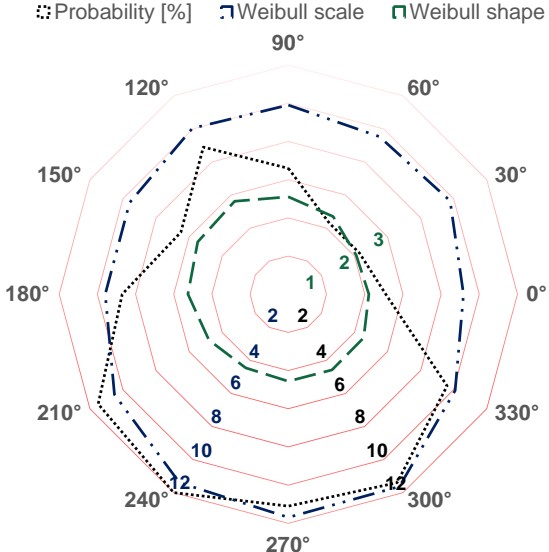

**Figure 8.** Site's wind rose.

The experiments have been carried out on an Intel Core i7-6600U CPU running at 2.50 GHz and with 16 GB of RAM.

Workflows for Task 1 in Fig. 4 and Fig. 5 have been implemented in the TOPFARM framework (DTU Wind Energy, 2018). The chosen MILP solver for Task 2 is the branch-and-cut solver implemented in IBM ILOG CPLEX Optimization Studio V12.10 IBM (2021).

### 3.1 Performance analysis of the cable layout algorithms

One of the main hypothesis for the effectiveness of Approach 2 (Fig. 5) is the accuracy of the estimation of the cable layout

cost by the heuristic of Fig. 2. The closer this cost estimation to the global optimization model (Sect. 2.2.2), the better the results of the simultaneous WT and cable layout design.

Aiming at performing a systematic evaluation of the heuristic algorithm, a pool of WTs layouts were generated trough the implementation of Approach 1 and Approach 2, by varying the computing time $t_r$ of the Random Search algorithm from few seconds up to $48 \, \text{h}$ ($\approx 210000$ function evaluations). Two different location criteria of the OSS are simulated for both

Approaches; first, at approximately the centroid of the WTs with possible correction as presented in Fig. 7, and second, at a designated location outside the OWF area.

It is important to bear in mind that the purpose of this section is not to conclude about the difference in the objective function value of both Approaches, but to focus on the difference of costs obtained from the heuristic and the global optimization model for the cable layout associated to each of the generated WTs layout. Intuitively, by utilizing Approach 1, a larger distance

between the WTs would be expected, maximizing the AEP, but the opposite is expected with Approach 2, as the cost of cables is taken into consideration. The stochasticity of the Random Search algorithm is advantageous for having a pool of realistic WTs layouts with high diversity within the previously described behaviour.




The performance analysis of the cable layout algorithms are graphed in Fig. 9 and in Fig. 10. For each WT layout from the generated pool, the average and standard deviation length between pairs of WTs are computed (in rotor diameters, $D$); 330 both measures provide a quantitative indication of how spread out the WTs are of each other within the OWF designated area: the greater the average and deviation length, the more separated and scattered they are. In blue color are displayed the results corresponding to WTs layouts generated after Approach 2, and in red color the ones after Approach 1. Triangle and circles are for OSS at approximately centroid, and at external location, respectively.

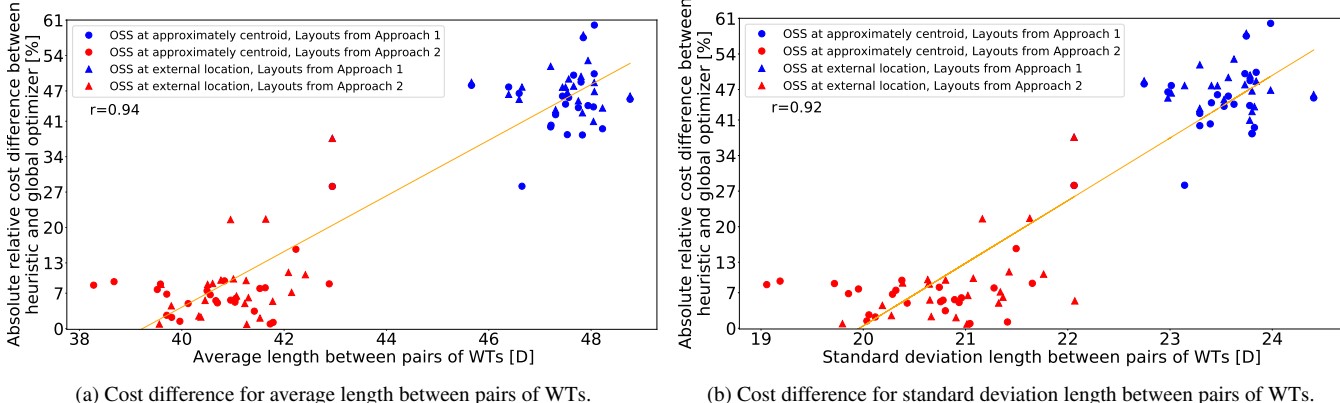

(a) Cost difference for average length between pairs of WTs.

(b) Cost difference for standard deviation length between pairs of WTs.

**Figure 9.** Cost comparison between cable layout algorithms: Absolute relative difference.

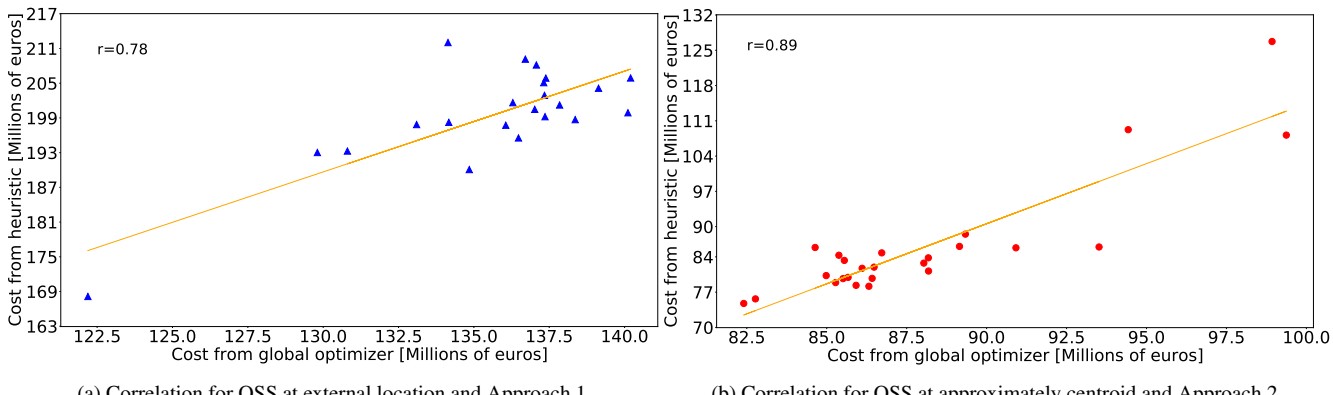

(a) Correlation for OSS at external location and Approach 1.

(b) Correlation for OSS at approximately centroid and Approach 2.

**Figure 10.** Cost comparison between cable layout algorithms: Pearson coefficient.

Fig. 9a shows the absolute relative cost difference between methods with respect to the global optimization model vs average 335 length, and Fig. 9b vs standard deviation length. The main takeaway is the confirmation of the initial expectation regarding the placement of WTs in each approach; there are clearly two clusters of points, for average length greater than $45D$ and standard deviation length greater than $22.7D$, the layouts from Approach 1 are placed, while for Approach 2 this is the case for values less than $43D$, and $22.1D$, respectively. Points at average $43D$ and standard deviation $22.1D$ correspond to $t_r \approx 0$, i.e., the





initial layout (see Fig. 6) which is in the center of the scale approximately. This outcome reflects the effect of including the

cable layout over the WT layout optimization process. It is also interesting to note that the clustering is independent of the OSS location. The second takeaway is the strong correlation between cost difference vs average length (Pearson coefficient, $r = 0.94$), and cost difference vs standard deviation length ($r = 0.92$). This means that the heuristic algorithm performs better at estimating for dense WTs layouts. The average absolute relative cost differences are 46% and 8% for Approach 1 and 2, respectively. This is a remarkably large difference in function of WTs layout characteristic.

From other perspective, Fig. 10 displays the correlation between the costs from the heuristic vs the global optimization model. Fig. 10b evidences the good but not so strong correlation for those layouts associated to Approach 2. In Fig. 10a the correlation is even worse for data obtained from Approach 1. According to these results, the heuristic algorithm leads to worse results in terms of cost difference and cost correlation for less dense WTs layouts. An ideal heuristic would have cost difference of 0%, correlation of 1, and very fast computing time (order of few hundreds of ms). The latter is confirmed, as for all presented

results, the heuristic takes between $100\,\mathrm{ms}$ to $300\,\mathrm{ms}$, while the exact method can go up to $10\,\mathrm{h}$ for gaps lower than 2%. More computing time (in average an order difference of hours to minutes) of the global optimization model is experienced for dense WTs layouts than for spread ones.

    The previous results suggest a compelling aspect: the initial layout impacts the overall optimization process, as the initial cost estimation from the heuristic would likely be inaccurate. The more spread in the initial layout, the greater the penalization

of the cable layout cost during the first iterations. The degree of impact of this deviation is yet unclear and hard to predict, but a seemingly good hypothesis is that a detriment of the final solution could be caused due to a potential sizeable reduction of the AEP. The absolute relative cost difference and correlation linked to the initial layout, and to the layouts towards denser arrangements, are deemed good enough for this application.

    To understand the underlying reason for the performance observed in Fig. 9 and in Fig. 10, the usage of available cables

of Table 3 by Approach 1 and Approach 2 (with OSS at approximately centroid) is shown in Fig. 11, where the average relative length differences with respect to the global optimization model are plotted for each cable type and approach. It can be noted that the heuristic designs cable layouts in Approach 1 using a disproportionately large amount of the most expensive cable (212% of type 3), even though the designed total length of cables is shorter with 14.59% on average. This trend is also noticeable for cable type 2 with an excess of 80.7%. The cheapest cable (type 1), on the contrary, is used less by the heuristic.

The exponential cost nature of cables (Eqn. 10) causes that this length disparity is reflected even more sharply monetarily. Notwithstanding this issue, for Approach 2 the most pronounced difference is shifted to cable type 2, but with the other two cables used less. The results for layouts from Approach 2 point out that a better balance of cables usage allows for a smaller cost difference and stronger correlation between both cable layout design methods. This evidence can be understood due to trade-off function of Algorithm 1 which makes that bigger clusters of WTs are formed when they are widely spread out, leading

to main feeders to support more WTs. More dense WTs layouts result in smaller groups of WTs, and consequently, more main feeders. This suggests that a WT-layout dependent trade-off cost function could bring along benefits for the performance of the heuristic algorithm.



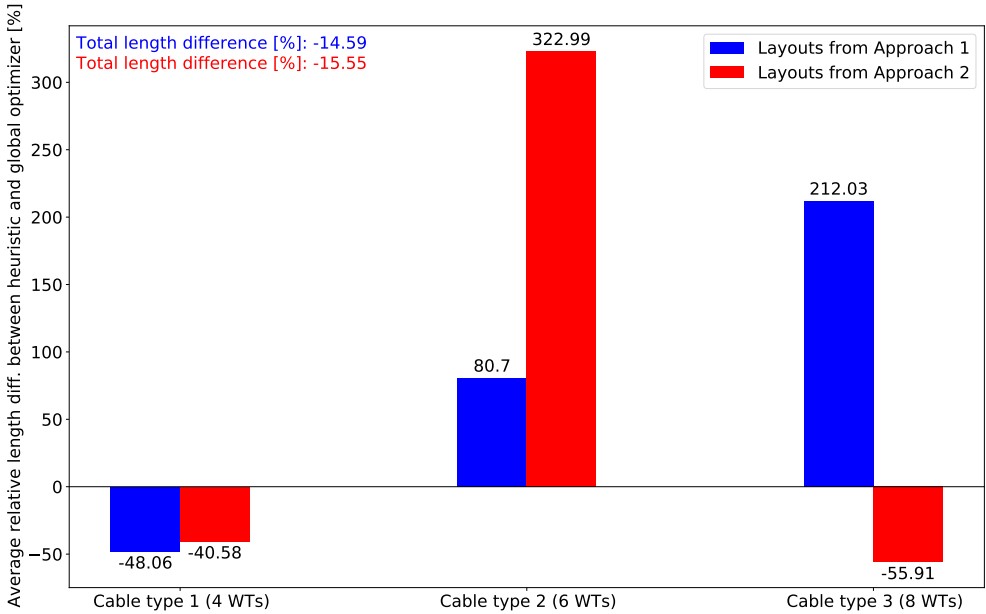

**Figure 11.** Cable length comparison between cable layout algorithms for OSS at approximately centroid.

## 3.2 Statistical analysis between Approach 1 and Approach 2

After a sensitivity analysis, the computing time for the statistical study between both approaches is fixed to $36\,\mathrm{h}$ ($\approx 170000$
function evaluations), valued regarded to be statistically sufficient to leverage the frameworks of Fig. 4 and Fig. 5 based on
numerous computational experiments. Inasmuch as the rather high computational time, eight runs of method in Fig. 6 are
executed for each Approach 1 and Approach 2 with both possible OSS locations.

The graphical representation of IRR for the four cases (approaches x OSS locations) through box-and-whisker diagrams
with inclusive median is displayed in Fig. 12. For the centroid OSS, all realizations from Approach 2 result in a better IRR
(Fig. 12a) than Approach 1, with an improvement between maximum values of 3.52%. Less dispersion in the data linked to
Approach 1 is noticed, with standard deviation of 0.011% vs a standard deviation of 0.020% in Approach 2. An interpretation
to this effect is that the heuristic introduces a high variability in function of the WT layout. The benefits from the simultaneous
approach are visible in Fig. 12b, with an IRR improvement between maximum values grows to 6% for the external OSS, also
presenting a larger variability than the data from the sequential approach (0.020% vs 0.015%). The larger improvement in the
case of a OSS externally located is attributed to the higher cost of the cable layout (see Fig. 10), increasing the weight of this
cost share into the overall economic metric (IRR). Approach 2 is hence able to exploit situations where submarine cables are
extensively required, for example with different OSS location, number of WTs, rated power, designated area, etc.





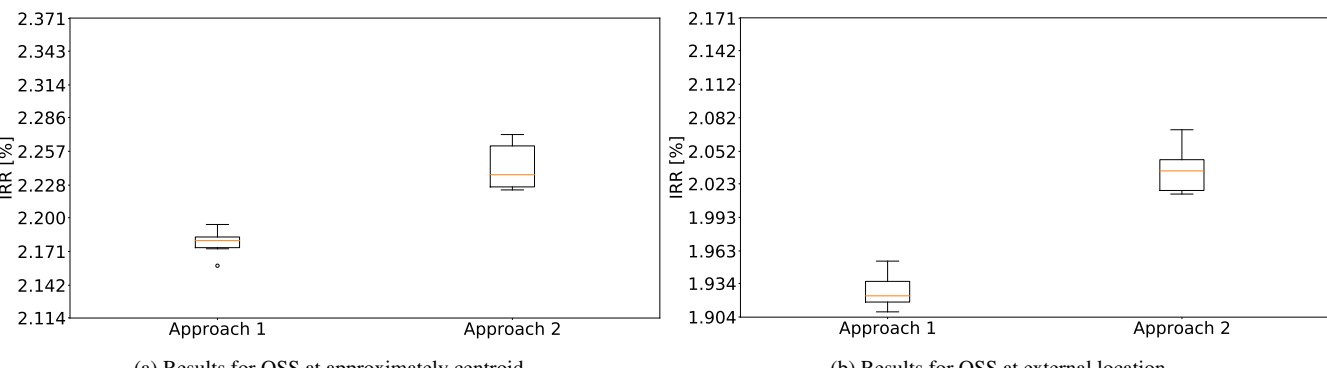

(a) Results for OSS at approximately centroid.

(b) Results for OSS at external location.

**Figure 12.** $IRR_o()$ comparison between Approach 1 and Approach 2.

The best layouts obtained by each approach - for centroid and external OSS respectively - are shown in Fig. 13 and Fig. 14. It is evident that the main difference between the pairs of OWFs is the concentration of WTs within the OWF designated area. 390 Note how both layouts from Approach 1 utilize more the area, with WTs highly located at the OWF borders, in contrast to those from Approach 2 where the WTs are closer to each other. The impact of the site's wind rose (Fig. 8) is also clear, as WTs are mostly aligned towards the prevailing wind direction (third quadrant of the coordinate plane).

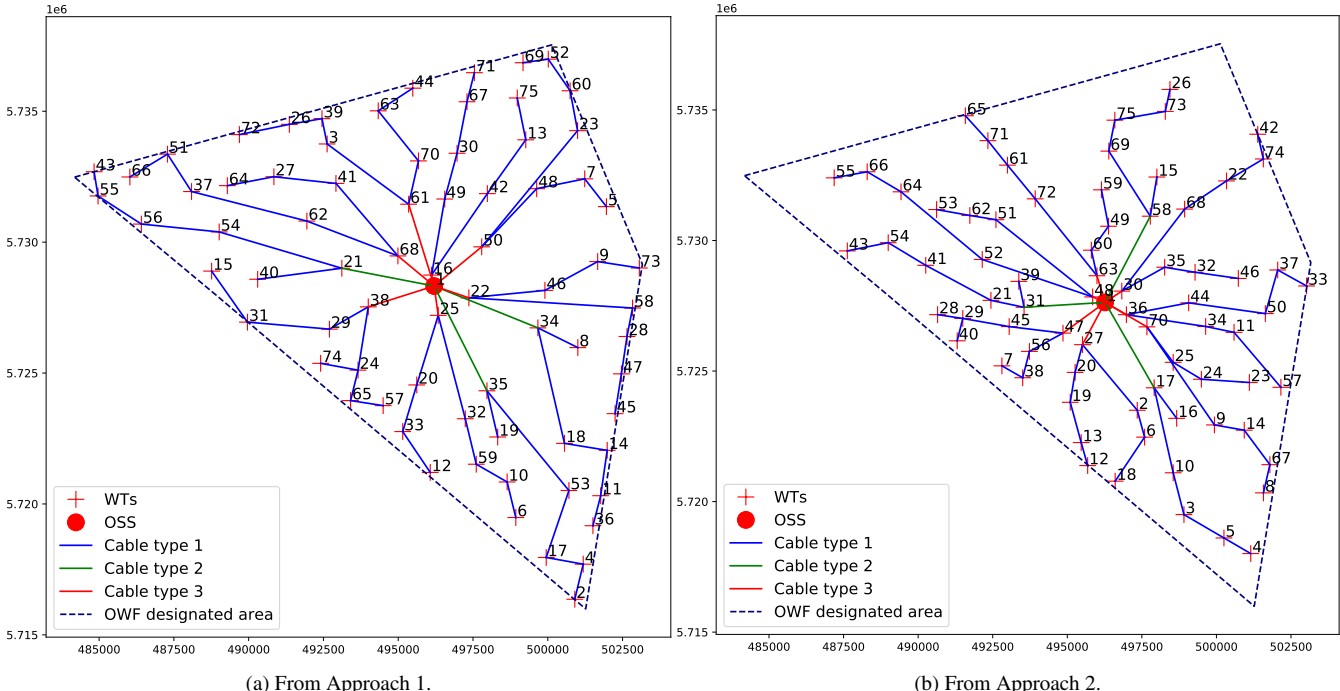

(a) From Approach 1.

(b) From Approach 2.

**Figure 13.** Designed WTs and cable layout with OSS at approximately the centroid.





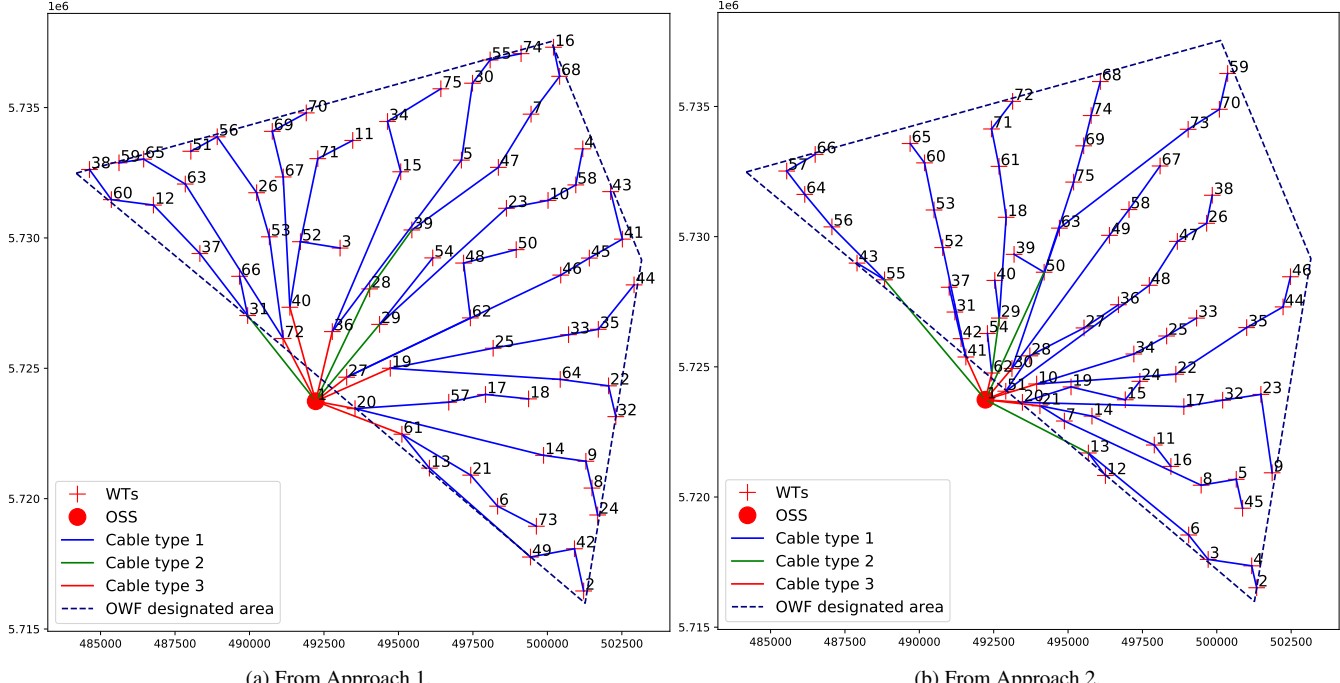

<table>
(a) From Approach 1.      (b) From Approach 2.
</table>

**Figure 14.** Designed WTs and cable layout with OSS at outside OWF limits.

Beyond the improvement over the overall IRR in random finite realizations, the contribution of the simultaneous approach should also be assessed from a statistical perspective to prove a significant impact, considering the stochasticity of the opti-

mization algorithm. A t-test for independent samples is conducted with this in mind. The main assumption to validly perform this parametric test is that the populations linked to each sample set are normally distributed. Although the probability distributions of both approaches are unknown and inaccessible, the normal distribution is a good default choice in this case of absence of prior knowledge about the form of the function (Goodfellow et al., 2016). Two reasons which back up this supposition are, first, the central limit theorem, and second, the fact that a normal distribution encodes the maximum amount of uncertainty

over the real numbers.

Table 4 and Table 5 summarize the results of the t-test assuming unequal variances for centroid and external OSS at approximately the centroid, and externally located, respectively. The null hypothesis is that the mean difference between feasible designs obtained from both approaches is equal to zero, i.e., Approach 2 does not improve Approach 1. Results are very clear, for both cases, the p-value is almost 0% (focus on the results of a one-tail distribution given that the directionality of average

comparison is clearly defined), meaning that there is a statistically significant difference between the two approaches, with the simultaneous one providing in average feasible designs located in a subset of the objective function's domain with greater quality than those stemmed from the sequential approach.

The results suggest two points. First, that the objective function defined in Task 1 of Fig. 5, $IRR_i()$, succeeds at having a good representation of the target function $IRR_o()$, due to the good performance of the heuristic for cable layout design. An





average absolute relative difference between $IRR_i()$ and $IRR_o()$ for several runs of Fig. 5 equal to 2.4% and 3.89% for OSS at centroid and outside, respectively, compared to the results of Fig. 4 of 50.69% and 70.41%, correspondingly, proves this aspect. Second, the ability of the simultaneous optimization framework to consistently find points which belong to a region of the target objective function with higher quality than the simultaneous counterpart.

**Table 4.** t-Test independent two-sample assuming unequal variances for OSS at approximately centroid

|  | *Approach 2 - $IRR_o()$* | *Approach 1 - $IRR_o()$* |
|---|---|---|
| Mean | 2,243246669 | 2,178664302 |
| Variance | 0,0003807 | 0,000106486 |
| Observations | 8 | 8 |
| Hypothesized Mean Difference | 0 | |
| df | 11 | |
| t Stat | 8,27582665 | |
| P(T>\|t Stat\|) one-tail (checked p-value) | 2,36281E-06 | |
| t Critical one-tail | 1,795884819 | |
| P(\|T\|>\|t Stat\|) two-tail | 4,72563E-06 | |
| t Critical two-tail | 2,20098516 | |

**Table 5.** t-Test independent two-sample assuming unequal variances for OSS at external location

|  | *Approach 2 - $IRR_o()$* | *Approach 1 - $IRR_o()$* |
|---|---|---|
| Mean | 2,035041596 | 1,926768153 |
| Variance | 0,000397414 | 0,000225622 |
| Observations | 8 | 8 |
| Hypothesized Mean Difference | 0 | |
| df | 13 | |
| t Stat | 12,26903326 | |
| P(T>\|t Stat\|) one-tail (checked p-value) | 8,00524E-09 | |
| t Critical one-tail | 1,770933396 | |
| P(\|T\|>\|t Stat\|) two-tail | 1,60105E-08 | |
| t Critical two-tail | 2,160368656 | |

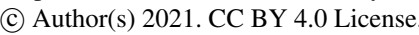

# 4   Conclusions

The proposed method provides an approach for simultaneous design of WTs and cable layout. The approach supports typical engineering constraints frequently used in this context, including, OWF designated area, minimum distance between WTs, tree topology for cable layout, thermal limits of available cables, maximum number of main feeders, and no crossing of cables.

The main novelties of this manuscript are: (i) Proposition of the simultaneous optimization framework, harmonizing different algorithms (Random Search, heuristic for the cable layout, and global optimization model) resulting in a formulation and
method with good computational properties, as tractability, efficiency, and effectiveness. (ii) Systematic performance assessment of the fast cost estimation from the cable heuristic compared to the global optimization model, identifying a connection between WTs layouts, and quality (relative difference, correlation, and computing time) of the estimation, which is considered fundamental for the full framework success. (iii) Rigorous benchmark of the proposed method vs the current practice using statistical tools.

The proposed simultaneous methodology has been applied to a large-scale offshore wind farm, demonstrating its feasibility and superiority compared to the sequential counterpart (Approach 1). The heuristic algorithm for the cable layout performs reasonably well for layouts arrangements with higher density, informing the Random Search algorithm of this cost share during the iterative optimization process. A sensitivity analysis for two different OSS locations (at approximately centroid and external of the OWF) points out the high correlation between the benefits of the simultaneous approach (Approach 2), and the overall
weight of the cable cost. For the OSS located at approximately the centroid, an improvement of the IRR of 3.52% is achieved, while for the external OSS, the improvement is boosted to 6%.

Finally, based on multiple runs of the optimization algorithm, it is possible to conclude that feasible points obtained after the Approach 2 are located in average in a region of the objective function's domain with higher quality than those from the Approach 1. This proves the effectiveness of the framework defined by Approach 2 in representing well the complete objective
function supporting both the WTs and cable layout (IRR), by means of the heuristic as a surrogate model, and at the same time, the capacity to obtain feasible points belonging to that domain region.

*Code availability.*   Available upon request.

*Data availability.*   Available upon request.

*Author contributions.*   JP designed the algorithms, approaches and case studies with inputs from NC. JP implemented the model and carried
out the numerical experiments. JP and NC analysed the results. JP prepared the manuscript with contributions from NC.





*Competing interests.*   No competing interests are present.

*Acknowledgements.*   Authors thank Dr. Katherine Dykes for her support in setting up TOPFARM simulations, general discussions and feedback on the manuscript, and Prof. Mathias Stolpe for the discussions and inputs shaping up the case studies.



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
