# Peer review of "A Framework for Simultaneous Design of Wind Turbines and Cable Layout in Offshore Wind"

_Wind Energy Science, 2021_

## Author Comment (AC1)

Dear Andrew P.J Stanley,

Thanks for taking your time to review our manuscript, for suggesting valuable changes for it, and for appreciating the contributions of our research. See changes after your comments in the new version of the article in Orange color. Find below responses to each of your comments:

**COMMENT 1:** First, in the literature review I think you should mention the paper "Wind plant system engineering through optimization of layout and yaw control" by Fleming et al. This paper focuses yaw control and layout optimization, but they make cabling considerations in the layout optimization.

**RESPONSE 1:** We have added the paper by Fleming et al. under the same category as the articles of Sanchez Perez-Moreno et al., Wade et al., and Amaral and Castro are classified. All these works include some considerations related to the effect of the cable length or investment, either as part of the objective function or as a constraint. However, they typically employ simple unconstrained heuristics (like Prim or Kruskal). In this paper, we take it further by proposing tailor-made heuristics that provide reasonably good cost estimation of the cable layout, which actually work satisfactorily in the context of simultaneous WT and cable layout optimization. The tailor-made heuristic supports for capacitated constraint to model maximum capacity, while ignoring other constraints to counteract the inherit suboptimal nature of heuristics. We also guarantee fully feasible designs taking into account the common engineering constraints of both problems.

**COMMENT 2:** Second, in the results I would like to see the correlation between AEP and cabling cost. There are several ways this could be presented, but I'm imagining a figure or figures like 9 and 10, which shows the AEP vs average length between wind turbines and/or AEP vs the costs from the global optimizer. I think this is important to show the reader and will help explain some of the results you talk about later.

**RESPONSE 2:** Thank you for the suggestion. Following the request, we produced the suggested graph (displayed below). Since the Pearson coefficient is 0.48, which represents a rather weak linear relationship, we are not sure if the plot would serve the purpose and transmit a clear message.

[Figure]

Instead, we chose to show the relationship between AEP and cabling cost as below. We think that this figure captures the relationship between AEP vs cable layout cost, transmitting a clearer message to the reader. The Figure is discussed as follows:

The observed differences of IRR between approaches presented in Fig. 12 are broken down in terms of AEP of cable layout cost in Fig. 13. Approach 1 consistently provides WT layouts with greater AEP than Approach 2 in both OSS locations, at expenses of more costly collection system networks; this illustrates the (most likely nonlinear) correlation between AEP vs cable layout cost. The robustness of the AEP maximization process when neglecting the cable layout (Approach 1) is evidenced by the rather low standard deviation (0.79 MW, approximately 0.02% of the average value). The previously elucidated variability of IRR in Approach 2 (Fig. 12) is also reflected in the AEP spread, which has a standard deviation of roughly 0.1% of the average value for both OSS locations. Oppositely, the spread of the cable layout cost is not as marked as for the AEP, yet a larger variation between Approach 1 and 2 is still observable.

[Figure]

(a) Results for OSS at approximately centroid.  (b) Results for OSS at external location.

**Figure 13.** Comparison of AEP and cable layout cost between Approach 1 and Approach 2.

**COMMENT 3:** Third, I think you should discuss some you expect the results to change as your parameters change. For example, as energy prices increase, you'd expect the optimizer to favor AEP more heavily, as distance from shore increases you would expect… The results for this paper are very interesting and sufficient to demonstrate your method, but I think this is important to discuss because you have only provided results for one set of parameters.

**RESPONSE 3:** Key parameters that affect the performance of Approach 2 are: energy price, cable costs, OSS location, and available area. Typical values for all of them have been used to set up the case study. A specific rigorous study could be conducted to quantify the effects of the variations of such inputs to the overall results. This is left for future works. The following paragraphs have been added to the manuscript.

At the end of Section 3:

The presented results are project-dependent, and therefore are vulnerable to variations of the set of input parameters. An increase in energy price would result in a heavier weighting of AEP during the optimization process, giving priority to a more spread out WT layout. The cable costs are another important parameter, that in turn if greater, could favor more strongly the cable layout design by bringing together the WTs; similar logic applies for OSS location further away to the WTs. The available area is also deemed as key, since larger areas would give more room to exploit trade-offs between WT and cable layouts when applying the design Approach 2.

At the end of the conclusions:

Typical values for the input parameters (energy price, cable costs, available area, OSS location, among others) are utilized to set up the case study. However, the outperforming capacity of Approach 2 over Approach 1 is affected by variations of those conditions: more expensive energy prices can result in a better performance of a sequential design approach, while greater cable costs could have the opposite effect. A more detailed analysis of the impact of those parameters over the results of the optimization could constitute future work.

**COMMENT 4:** Line 245: "Subsequently, the IRR metric may weight out more the AEP…" – This sounds off. Probably a good idea to reword.

**RESPONSE 4:** This has been changed to:

Therefore, IRR could implicitly favor AEP (cash flow) over cable cost (CAPEX)

**COMMENT 5:** Line 282: I don't think this necessarily guarantees there is no overlap between turbines and the OSS. What if the 4 turbines nearest the centroid were not arranged in a square? Like, a triangle with one in the middle but still meeting minimum spacing constraints?

**RESPONSE 5:** Thanks for the observation. This is true. What we mean is that the proposed approach is a simple heuristic that helps overcome the described situation of overlapping. It is definitely not a bulletproof concept, but it gives good results in fast computing time. The paragraph has been changed to:

Since, in theory, the location of the OSS, $\{x_1, y_1\}$, can take infinite values, a course of action is to fix the OSS location in the centroid of the WTs. However, only considering the project area centroid could lead to unrealistic designs, as presented in Fig. 7, where an overlapping between a WT and the OSS appears. In order to decrease the occurrence likelihood of this erratic design, a simple heuristic rule for the OSS location is presented in Fig. 7. The OSS is displaced to the centroid of the four nearest WTs, in case of the original distance between the points denoted by the OSS and nearest WT is under a specific threshold. For the particular case of a square disposition of WTs, assuming a minimum distance between WTs of 2D, a minimum distance between OSS and WT of √2D is ensured after this correction. This heuristic does not guarantee success in respecting the distance threshold for any WTs arrangement; however, it represents a good compromise between effectiveness and computing burden.

**COMMENT 6:** Figure 8 (or anywhere talking about the resource): It's worth mentioning the number of wind speed bins you used

**RESPONSE 6:** Thank you for the suggestion. This information has been added to the caption of Figure 8.

**COMMENT 7:** Line 322: "trough" – typo

**RESPONSE 7:** Thanks for the observation. We have corrected this typo.

**COMMENT 8:** "The main takeaway…" - I would reword this sentence, it's hard to follow.

**RESPONSE 8:** Thanks for the observation. We have changed this sentence by:

"This figure confirms the initial expectation regarding the placement…"

---

## Author Comment (AC2)

Dear Michael Muskulus,

Thanks for taking your time to review our manuscript, for suggesting valuable changes for it, and for appreciating the contributions of our research. See changes after your comments in the new version of the article in Purple color. Find below responses to each of your comments:

**COMMENT 1:** My main technical question is actually on the (previously published) formulation of the cable layout optimization problem as a MILP. Is this not much too complicated? Why is it not sufficient to only use the connectivity variables x_ij and optimize their values, under suitable constraints? It seems that the objective function and all constraints can be expressed as functions of the x_ij. Or am I mistaken? And what is the role of the f() function? Why are the y_ij^k variables needed? What is the "maximum number of WTs connectable through an arc"? The MILP formulation has to be better explained!

**RESPONSE 1:** $y_{ij}^k$ is the main variable in the model. This variable models the decision of how many $k$ wind turbines are connected by the arc (i,j). $x_{ij}$ is an auxiliary variable that allows for a more intuitive and compact full representation of the model, as it represents which arc $(i, j)$ is active. Our hypothesis is that as both variables are linked through Equation (8), this may help tighten up the model and work as valid inequalities. The objective function therefore has to be expressed in function of variables $y_{ij}^k$, which encode the cost value of the optimum cable to support those $k$ wind turbines, computed in a preprocessing stage. The function $f()$ allows reduction of variables number as it computes the maximum number of wind turbines connectable through arc $(i, j)$, so for example, only arcs that are connecting directly to the OSS can support the maximum number of WTs $U$, which is not true for arcs connecting to a wind turbine supporting maximum $U - 1$ turbines.

Some sentences have been added to the model description as presented in lines 215 to 220, as follows:

Eq. (2) defines the variables of the model. Binary variable xij is one if the arc (i,j) is selected in the solution, and zero otherwise. Likewise, binary variable ykij models the k number of WTs supported upstream (with respect to flow towards the OSS) from j, including the WT at node j (under the condition that xij = 1). Function f(i) , that allows for variables number reduction, maps from tail node i to maximum number of WTs connectable km through an arc (i,j). If i = 1 (i.e. the OSS), then km = f(i) = U, otherwise km = f(i) = U −1. Eq. (3) is the objective function. Cost parameter ckij encodes the optimum cost to connect k WTs through arc (i,j) and is obtained similarly to Step 2 in Fig. 2. [C2] is enforced at this point as well. Eq. (4) ensures simultaneously a tree topology, only one cable type used per arc, and the head (j)-tail (i) convention, while Eq. (5) is the flow conservation which avoids forest graph; both Eq. (4) and Eq. (5) guarantee [C1]. Eq. (6) expresses the requirement of [C3]. The set χ stores pairs of arcs {(i,j),(u,v)}, which are crossing each other. Excluding crossing arcs ([C4]) in the solution is ensured by the simultaneous application of Eq. (7) and Eq. (8). Finally, Eq. (9) defines a set of valid inequalities to tighten the mathematical model.

**COMMENT 2:** Optimization of wind turbine positions is either using no cable layout cost, or a fast, heuristic approach. Is it not possible to run the mixed integer linear programming (MILP) cable optimization during this optimization? Is it so slow? This would provide reference results against which the results of both approximations could be compared to.

**RESPONSE 2:** As stated in line 355 of the manuscript, the exact solver for the MILP model can take up to 10 hours and no less than dozens of minutes (in rare cases) to terminate given a minimum gap of 2%. Therefore, it is unpractical to run this solver during the simultaneous optimization. We rely on the presented heuristics as a proxy to estimate the cable layout cost.

**COMMENT 3:** The constraint C1 enforcing a tree topology: Why are no back-connections considered, offering a redundant electrical path in case of a fault at one turbine? Of course this would complicate the optimization problem and its formulation, but why are these technical solutions excluded a priori? The layout cost would be higher, of course, but OPEX or financial risk might be lower due to the redundancy.

**RESPONSE 3:** This manuscript addresses the collection system problem in OWFs considering only radial designs. First, this is because a majority of OWFs developers only consider networks with this topology type. Second, cable loops can be studied with accentuated focus in a subsequent stage, where after reliability analysis, redundant cables can be installed in the radial system, in order to account for contingencies of cables and wind turbines. In the operating mode, OWFs generally maintain the switchgears of those redundant cables in open circuit, maneuvering these components after a fault in an electrical component has been identified. Finally, we have conducted studies regarding reliability-based optimization (see article "Reliability-based Topology Optimization for Offshore Wind Farm Collection System", https://onlinelibrary.wiley.com/doi/full/10.1002/we.2660). Results indicate that the benefits to design redundant topologies depend on parameters such as Mean Time Between Failures (MTBF), and Mean Time To Repair (MTTR), with rather limited set of reasonable available data.

**COMMENT 4:** page 7: Implementation of constraints. The approach only works since the methods in Step 1 are all greedy algorithms, right? And even then there are problems, as the authors mention (infeasible points can be obtained), unless a better approach is found. Simply relaxing the constraints (not considering C3 and C4, for example) is a somewhat radical solution. But since this has been done, it would be interesting to know how often the cable layout was infeasible before running the MILP for it.

**RESPONSE 4:** The presented greedy heuristics are very fast, however they have issues of scalability in terms of supporting a large number of constraints. We have chosen to respect constraints C1 and C2, as they are rather straight-forward to be included by the heuristics. The benefits of relaxing constraints C3 and C4 are twofold. First, computing time is saved trying to satisfy them, process that can be cumbersome (see article of the first author, https://arxiv.org/abs/2108.13973). Second, as discussed in the manuscript, by relaxing the constraints a counteracting effect is theoretically achieved, as heuristics inherently provide suboptimal solutions when fully satisfying all constraints. It is out of interested in this manuscript to actually present the satisfiability of the constraints for each WT layout. Instead, the reflection over cost is presented in Figures 9 and 10, where these are compared to the solution of the exact optimizer.

**COMMENT 5:** Algorithm 1: needs comments! Otherwise this is hard to read. And what is the inverse of an arc and why is it needed? (Why are directed graphs considered here, not undirected ones?) What are its inputs? What idea does the tradeoff value T represent intuitively?

**RESPONSE 5:** Thanks for the observation, we have tried to improve the description of Algorithm 1, please see the new comments in lines 180 to 190 of the manuscript. The definition or an arc ($a = (i,j)$) and its inverse ($\bar{a} = (j,i)$) are clarified in the manuscript. Because the directionality of arcs matter, the input must be given in form of a directed graph. The directionality has a clear influence over the trade-off values. The trade-off values represent the balance between the length of an arc and the estimated distance to the OSS; therefore, priority is given to connect WTs located the farthest to the OSS for a given length.

New description:

The input of the algorithm are collected in the weighted directed graph. The first five lines initialize the useful sets. The most180 important is the trade-off set To using the weight parameters pi; note that due to the the weight parameter definition an arc a = (i,j) and its inverse ⁻a = (j,i) must be considered independently. The iteration process starts at line six and continues until a fully connected tree graph is obtained. In each iteration the arc with the lowest trade-off value is incorporated to the tree, as 7 long as it satisfies [C1] and [C2] (lines seven to nine). According to the trade-off value, priority is given to the arcs located the farthest to the OSS for the same arc length, as the greater pi, the lower ta. In case constraints are met, the component sets (that include nodes i and j of the selected arc ao) are merged, and the trade-off values linked to the newly formed component are updated (lines 10 to 14). Otherwise, the arc ao and its inverse ⁻ao become completely banned from the design process, by equalizing their trade-off values to infinite (lines 16 to 17). The indirect graph Gd in line 19 contains the graph tree nullifying any directionality of arcs.

**COMMENT 6:** page 7: Unclear notation. What is the set "B" in the definition of the "trivial optimization problem"? And what is "X_c"?

**RESPONSE 6:** Thanks for the observation, the text of lines 190 to 195 has been updated to:

Let β be the number of WTs connected through edge e ∈Eo with length de. The following trivial optimization problem must be solved: min {X$^T$c ·C ·de : X$^T$c ·U ≤β,‖Xc‖1 = 1,Xc ∈B|T|}, being B|T| the |T|-tuple binary set. The solution provides the cheapest cable t ∈T able to support β WTs via binary variable xct ∈Xc, where each tuple (e ∈Eo,β) defines an independent problem solved in linear running time.

**COMMENT 7:** Fig. 6: this algorithm could be explained better, e.g. with fewer formulas

**RESPONSE 7:** We consider that the formulas help to formalize the operating principles of the algorithm. The text from lines 296 to 303 complement the Fig. 6. Additionally, the reference (Feng and Shen, 2015) could be read for more details of the algorithm.

**COMMENT 8:** What about uncertainty (e.g. in the spot price) - how would that influence the optimization results?

**RESPONSE 8:** The following paragraphs discuss the influence of input parameters variability:

At the end of Section 3:

The presented results are project-dependent, and therefore are vulnerable to variations of the set of input parameters. An increase in energy price would result in a heavier weighting of AEP during the optimization process, giving priority to a more spread out WT layout. The cable costs are another important parameter, that in turn if greater, could favor more strongly the cable layout design by bringing together the WTs; similar logic applies for OSS location further away to the WTs. The available area is also deemed as key, since larger areas would give more room to exploit trade-offs between WT and cable layouts when applying the design Approach 2.

At the end of the conclusions:

Typical values for the input parameters (energy price, cable costs, available area, OSS location, among others) are utilized to set up the case study. However, the outperforming capacity of Approach 2 over Approach 1 is affected by variations of those conditions: more expensive energy prices can result in a better performance of a sequential design approach, while greater cable costs could have the opposite effect. A more detailed analysis of the impact of those parameters over the results of the optimization could constitute future work.

**COMMENT 9:** Table 3: I assume that U is given in terms of number of turbines supported by the respective cable? Or what are its units?

**RESPONSE 9:** Thanks for the observation. Yes, it is the number of wind turbines. This description has been added to Table 3.

**COMMENT 10:** page 16: Unclear notation. "correspond to t_r \approx 0" - What is t_r?

**RESPONSE 10:** Thanks for the observation. This is the computing time of the Random Search algorithm. This has been clarified in the text as:

tr ≈0 (computing time of the random search algorithm, see Fig. 6

**COMMENT 11:** Unclear notation: What is IRR_i and what is IRR_0?

**RESPONSE 11:** Both variables are defined in Figure 4 and 5. They are the IRR value before and after the Task 2, respectively.

**COMMENT 12:** Table 4: Do not use komma, but decimal point. Too many decimals given. Why are results for both one-tailed and two-tailed tests given?

**RESPONSE 12:** Thanks for the observation. Both Tables have been corrected correspondingly.

**COMMENT 13:** Conclusions: "an improvement of the IRR of 3.52% percent is achieved..." - This is when comparing the sequential and simultaneous heuristic approaches, I assume? More interesting would be, as already mentioned above, how large the difference with the optimum (as derived by MILP) would be between the heuristic-derived and the MILP layout.

**RESPONSE 13:** We have added the following sentence to the conclusion:

The heuristic algorithm for the cable layout performs reasonably well for layouts arrangements with higher density, with an average absolute relative cost difference with respect to the global optimization model of 8%.

**COMMENT 14:** "Code available upon request": Nowadays, statements like that are hard to justify, as it is so easy to upload the code to an institutional or other repository (e.g. Github).

**RESPONSE 14:** Thanks for the recommendation. A GitHub repo has been created for the code.

**COMMENT 15:** Data available upon request: This is marginally acceptable, but the data should ideally be uploaded into a public repository (e.g. Zenodo).

**RESPONSE 15:** Thanks for the recommendation. A Zenodo repo has been created for the data.

**COMMENT 16:** page 5: "wind directional bits" should probably be "... bins"?

**RESPONSE 16:** Thanks for noticing the typo. We have fixed it.

**COMMENT 17:** page 21: "with higher quality than the simultaneous counterpart" - this should probably be "... than the sequential counterpart"?

**RESPONSE 17:** Thanks for noticing the typo. We have fixed it.